# Contraceptive uptake and characteristics in the immediate postpartum period among women attending Tygerberg Academic Hospital

Denise Sproul[1,2]*, Michael McCaul[1], Gerhard Hanekom[2]

**1** Divison of Epidemiology and Biostatistics, Department of Global Health, Stellenbosch University, Cape Town, South Africa, **2** Department of Obstetrics and Gynaecology, Stellenbosch University, Cape Town, South Africa

* dsproul@sun.ac.za

## Abstract

The immediate postpartum period is a unique opportunity to motivate and educate women on family planning options. This study aimed to determine the prevalence of family planning use in our setting. Secondary objectives include determining the frequency of the various methods used, the safety of the method(s), common risk factors, the interval spacing between births and associations with family planning method choices. A cross-sectional study was conducted at Tygerberg Academic Hospital, South Africa, which serves a predominantly low socioeconomic status and high-risk obstetric population. Medical records were assessed prospectively from January 2023 till July 2023. Data was collected and captured from maternity medical records on the day of discharge. The statistical analysis was done using frequencies and percentages, with 95% confidence intervals (95%CI). A secondary bivariant analysis was done to evaluate the associations between these variables. Of the 1066 participants included, 97.5% (95%CI 96.5–98.5) received family planning. The majority of the patients (48.8%) requested the injectable depo medroxyprogesterone acetate. These patients who chose this method were on average younger (OR 0.962, p < 0.001), had a lower BMI (OR 0.983, p = 0.012), and were a higher parity (OR 0.721, p < 0.001). HIV positive women were also 35% less likely (OR 0.651, p = 0.008) to choose this method compared to other methods. Of the 1039 patients who had received family planning, 47.8% were classified into a World Health Organization medical eligibility category three. This was almost exclusively attributed due to patients who were breastfeeding and requested the depo medroxyprogesterone acetate as their family planning method postpartum. The majority (66%) of the patients had spaced their pregnancies more than two years apart. The exceptionally high prevalence of uptake of family planning indicates it can be successfully implemented in the postpartum period. Patients should be counselled individually on the safest and the most appropriate method.

**Data availability statement:** All relevant data are within the paper and its Supporting Information files (Excel spread sheet, tables and interpretation).

**Funding:** We would like to acknowledge the Temporary Research funding support from Stellenbosch University's Subcommittee C of the Research Committee. As well as the funding received from the GELA project which is part of the EDCTP2 programme supported by the European Union (grant number RI2020S-3303 - GELA). The views and opinions of authors expressed herein do not necessarily state or reflect those of EDCTP. The funders had no role in study design, data collection and analysis, decision to publish, or preparation of the manuscript.

**Competing interests:** The authors have declared that no competing interests exist.

## Introduction

Globally, family planning is recognized as a life-saving intervention [1]. Family planning is an effective method to reduce unintended pregnancies and live birth rates, thus reducing maternal mortality rates. It can also have an influence on education, promotion of gender equality and women's rights, and can assist in reduction of human immunodeficiency virus (HIV) by use of barrier contraception in the form of male and female condoms [2]. This unique intervention could prevent a third of maternal deaths by delaying motherhood, allowing for appropriate interval spacing between births, limiting unplanned and unwanted pregnancies and subsequent abortions. For these reasons, the World Health Organization (WHO) issued a report in 2019 recommending family planning as one of the critical strategies in reducing maternal mortality [2,3]. Another recommendation is to space birth intervals by at least 24 months after childbirth to reduce the risk of adverse maternal and neonatal outcomes [4]. Evidence has shown that spacing live births by contraception can save more than two million newborns annually [2].

The immediate postpartum period is a unique opportunity to motivate and educate women on family planning options and provide a suitable method for each patient, individualizing her options to best suit her needs and fertility desires [5]. There are few studies that describe family planning uptake in the postpartum period, and even less in South Africa [6,7]. The rate of unintended pregnancies is estimated to be as high as 50% in Sub-Saharan Africa [8].

This study seeks to explore the uptake of postpartum family planning within our setting. By examining patterns of contraceptive use after postpartum, we could have a better understanding of how current service delivery aligns with recommendations from the World Health Organization (WHO) and the Royal College of Obstetrics and Gynaecology (RCOG), which advocate for comprehensive family planning access among reproductive-age women. These recommendations are based on evidence suggesting that effective postpartum contraception may help reduce unintended pregnancies, elective terminations, and promote healthier spacing between births. [1,3–5,9,10].

Secondary objectives included assessing the frequency of the different methods of contraception, determining the safety of the methods received by using the WHO medical eligibility category (MEC) scoring system to describe risk factors and co-morbidities of our population and to describe the spacing trends between previous and index pregnancy [11].

## Methods

### Study population and sample

We conducted a descriptive quantitative study at Tygerberg Academic Hospital, a tertiary referral centre in Cape Town, South Africa. A prospective data collection took place from January 2023 to July 2023, until a sample size of 1066 participants, according to the sample size calculation as described below, was met. The Hospital is a public, academic institution that serves a predominantly low socioeconomic

status (SES) and high-risk obstetric population, of majority African and mixed racial groups. Ethical approval was obtained from the Health Research Ethics Committee (HREC) of Stellenbosch University (S22/09/188), as well as the Western Cape Government for access to records and patient files. Informed consent was waivered, as only medical records were used for data collection with no patient interaction; this waiver was supported and approved by the HREC.

Participants aged older than 18 years and who were managed in the postpartum period at Tygerberg Academic Hospital were included into the study, regardless of whether the pregnancy ended in a live birth, a stillbirth, or an intrapartum death. Postpartum patients who required longer hospital admission and transfer to the gynecology wards were excluded from the study.

Participants were identified by their discharge records and medical records at the medical clerks in the postnatal wards, on the day of discharge which is usually two- or three-days post-delivery. Maternity records completed by health care professionals during their admission, together with their completed discharge forms were used for data collection and capture. Outstanding laboratory results were searched for on the online laboratory system and follow-up appointments were checked using the hospital's patient electronic record system. Data were anonymized to ensure privacy and confidentiality of participants' personal information, and each participant was assigned a unique identifier.

## Measures

To determine the uptake of family planning in the immediate postpartum period, data was collected on whether the patient had received the family planning method of their choice before discharge. This is marked on the discharge summary as well as the prescription chart. Other information recorded included which method was received, the patients risk factors (such as comorbidities and body mass index), baseline characteristics (including age, gravidity, parity, booking biochemistry), pregnancy complications, birth outcomes (live birth or stillbirth) and method of feeding (breastfeeding or formula feeding). Other measures investigated was the WHO Medical eligibility criteria score and the interval spacing between live births. Information biases were limited by the capturing of medical notes done by the same two research assistants to keep collection and capturing methods consistent.

## Statistical analysis

For the primary outcome of uptake of family planning in the immediate postpartum period, we calculated a sample size of 1066 anticipating a frequency of 50%, using a 95% confidence interval (95% CI) using OpenEpi sample size calculator [12]. The 50% frequency was chosen as the most conservative estimate maximizing the sample size. Stata version 17.0 was used for the data analysis (StataCorp, Texas 2021). For assessing the uptake of family planning in postpartum women, we calculated frequency and percentages, with a 95% confident interval (95% CI). Demographics of participants are described using frequency tables and summarized using mean, standard deviation (SD), and range for continuous variables if normally distributed, and median and interquartile ranges for non-normally distributed data. For the secondary objectives, we used frequency tables and graphs, and 95% confident intervals, to summarize data. A bivariant and multivariant secondary analysis was conducted to assess associations between variables and choice of family planning methods.

## Results

Over a period of seven months 1066 participants were recruited for this study and had a recorded family planning method of their choice. The uptake of family planning in postpartum patients in our setting was 97.5% (95% CI 96.3–98.3) (Fig 1).

The maternal characteristics are described in Table 1. The proportion of patients found to be syphilis negative was 96.5%. Of the 196 (18.4%) patients who were HIV positive, the majority (85.7%) had viral loads (VL) of less than 100 copies, indicating viral suppression. In terms of exposure to illegal drugs and alcohol during pregnancy 16.7% of patients admitted to smoking, 1.5% to using illicit drugs and 12.1% to the use of alcohol.

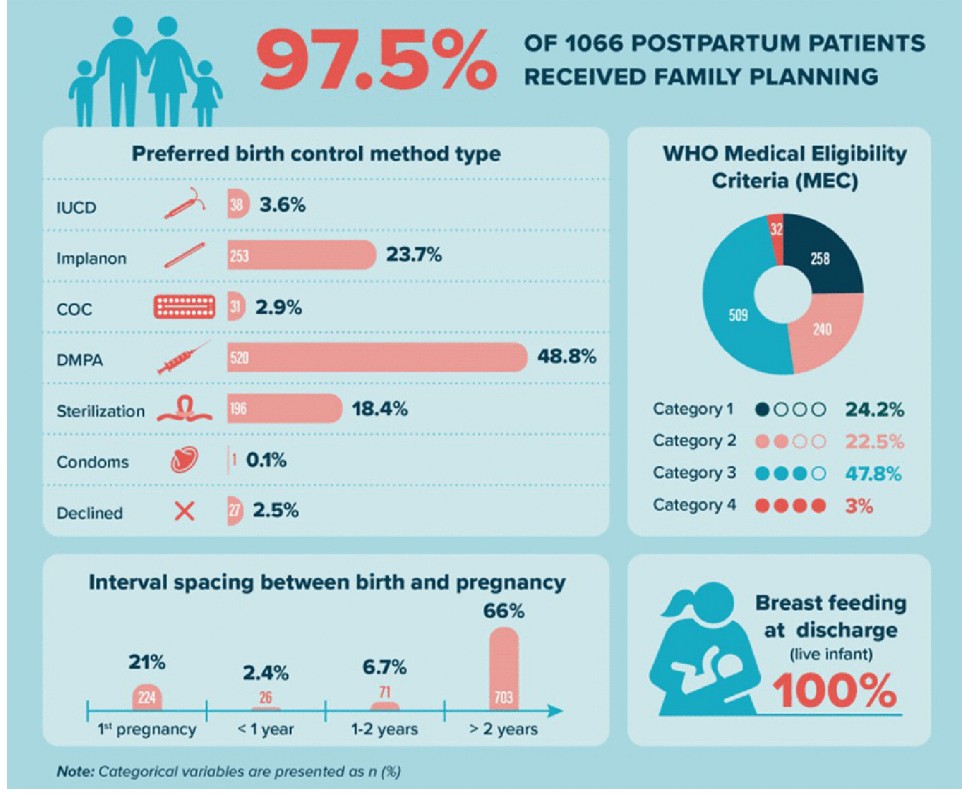

**Fig 1. Description of family planning amongst postpartum patients at Tygerberg Academic Hospital.**

The most common risk factors amongst the sample population were advanced maternal age (26.2%), defined as older then of 35 years of age, a high body mass index (BMI) above 30 kg/m² (58%), chronic hypertension (CHPT) (10.9%), and diabetes mellitus type two (3.5%).

The delivery characteristics are described in Table 2. The proportion of participants who had delivered via cesarean section was 49.3% compared to 50.7% who had undergone a normal vaginal delivery (NVD). All the women whose pregnancy resulted in a live birth (1036; 97.2%) were recorded as breastfeeding their neonate at the time of discharge (Fig 1).

The majority of the patients requested the three-monthly progesterone only injectable depo medroxyprogesterone acetate (DMPA) (48.8%). The remainder received the progesterone implant (23.7%) and sterilization by means of tubal ligation (18.4%) and a minority of patients requested the combined oral contraceptives (COC) (2.9%), condoms (0.1%) or the copper intrauterine device (IUCD) (3.6%). Despite receiving individual counselling, a minority (2.5%) of patients declined receiving family planning on discharge (Fig 1).

Using the WHO Medical Eligibility criteria (MEC), Fig 2, guidance on safe family planning prescription, each patient was analyzed according to their individual risk factors and placed into one of the four categories. Of the 1039 patients who had received family planning 47.8% of patients were classified into a MEC category three, 24.2% in MEC category one, 22.5% into MEC category two and only 3% into MEC category four (Fig 1). We found that 66% of the patients had spaced their pregnancies more than two years apart (Fig 1).

Table 3 presents the unadjusted (crude) associations between various factors and the choice of family planning methods comparing DMPA to the composite other methods. Women using DMPA are younger on average (29.4 years) compared to those using other methods (30.9 years, OR 0.962, p<0.001). Women who chose DMPA have a slightly lower

**Table 1. Maternal baseline characteristics of postpartum participants at Tygerberg Academic Hospital between January and July 2023.**

| Baseline characteristics (n) | Mean (IQR) |
|---|---|
| Age (years) | 30 (25– 35) |
| Gravidity | 3 (2–4) |
| Parity | 2 (2–3) |
| Body Mass Index (kg/m²) | 32 (26– 40) |
| **Maternal booking characteristics** | **n (%)** |
| Syphilis (RPR) positive | 39 (3.5) |
| HIV + | 196 (18.4) |
| Viral Load < 100 copies | 168 (85.8) |
| Viral Load > 100 copies | 28 (2.6) |
| Smoking (yes/no) | 176 (16.7) |
| Illicit drug use | 16 (1.5) |
| Alcohol use | 127 (12.1) |
| **Risk factors (n)** | n (%) |
| Advanced maternal age | 279 (26.2) |
| BMI > 30 kg/m² | 594 (58.6) |
| BMI > 40 kg/m² | 281 (27.7) |
| Chronic hypertension (cHPT) | 116 (10.9) |
| Diabetes mellitus type 2 | 37 (3.5) |

**Table 2. Delivery characteristics of participants at Tygerberg Academic Hospital between January and July 2023.**

| Mode of delivery | n (%) |
|---|---|
| Normal Vaginal Delivery | 540 (50.7) |
| Cesarean Section | 526 (49.3) |
| **Birth Outcomes** | **n (%)** |
| Live Birth | 1036 (97.2) |
| Intrauterine Death | 30 (2.8) |
| **Estimated Gestational Age at delivery (weeks)** | 38 (37-40) |

BMI than those using other methods (OR 0.983, $p = 0.012$). HIV-positive women are less likely to use DMPA (OR = 0.651, $p = 0.008$). Higher parity is associated with a lower likelihood of using Depo (OR = 0.721, $p < 0.001$), suggesting that women with more children prefer other family planning methods.

Table 4 shows the adjusted associations after controlling for other factors. Age and HIV status are no longer significant factors for choosing DMPA over other methods. Parity (OR 0.689, $p < 0.001$) remains statistically significant, indicating that even after adjustment, higher parity decreases the likelihood of choosing DMPA. BMI is a borderline significant (OR 0.986, $p = 0.053$), suggesting that BMI may still play a role in DMPA use.

## Discussion

Our study found a 97.5% uptake of family planning in the immediate postpartum period of high-risk obstetrics patients in a low SES. This finding indicates a much higher uptake rate than other studies reported in South Africa. A study conducted in the rural area of Mpumalanga, South Africa, 2021 by Mandell et al. showed a prevalence of 70.9% family planning use in postpartum women [6]. Whereas uptake of family planning at a hospital in an urban area in Gauteng Province, South

| | | Definition | Clinical Judgement | With limited clinical judgement |
|---|---|---|---|---|
| Category 1 | ●○○○ | A condition for which there is no restriction for the use of the contraceptive method | Use method in any circumstance | Yes, use the method |
| Category 2 | ●●○○ | A condition where the advantages of using the method generally outweigh the theoretical or prove risks | Generally, use method | |
| Category 3 | ●●●○ | A condition where the theoretical or proven risks usually outweigh the advantages of using the method | Use of the method not usually recommended unless other more appropriate methods are not available or not acceptable | No, do not use the method |
| Category 4 | ●●●● | A condition that represents an unacceptable health risk if the contraceptive method is used | Method not to be used | |

**Fig 2. WHO medical eligibility criteria for safe use of family planning.**

**Table 3. Bivariate analysis to determine associations between DMPA and other methods to maternal variables.**

| Variable | Family Plan Methods | | Odds Ratio | P-Value | 95% CI |
|---|---|---|---|---|---|
| | **DMPA** | **Other** | | | |
| Age | 29.4 (6.4) | 30.9 (6.1) | 0.962 | <0.001 | (0.94, 0.98) |
| Parity | 29.4 (6.4) | 30.9 (6.1) | 0.721 | <0.001 | (0.652, 0.797) |
| BMI | 32.3(9.3) | 33.8(9.3) | 0.983 | 0.012 | (0.98, 0.99) |
| **HIV Status** | | | | | |
| Positive | 80 (15.4%) | 113(21.8%) | 0.651 | 0.008 | (0.475, 0.894) |
| Negative | 440 (84.62%) | 405(78.19%) | 1 | | |
| **CHPT** | | | | | |
| Yes | 51 (9.82%) | 57 (10.98%) | 0.881 | 0.535 | (0.591, 1.314) |
| No | 469 (90.19%) | 462 (89.02%) | 1 | | |

**Table 4. Multivariate analysis describing adjusted associations after controlling for other factors, between variables and DMPA compared to other family planning choices.**

| Variable | Odds Ratio | 95% CI | P-Value |
| --- | --- | --- | --- |
| Age | 1.007 | (0.980 - 1.033) | 0.626 |
| HIV Status (Positive) | 0.784 | (0.558 - 1.101) | 0.161 |
| Parity | 0.689 | (0.602 - 0.784) | < 0.001 |
| BMI | 0.986 | (0.97 - 1.00) | 0.053 |

Africa was estimated at 55.3% [7]. This high uptake of family planning is aligned with the recommendations of the WHO and RCOG, which advocate for all postpartum women to receive family planning as a strategy to reduce maternity and neonatal morbidity and mortality rates, prevent unwanted pregnancies and abortions, and allow adequate interval spacing between births to reduce complications in the next pregnancy [5,9].

The high contraceptive uptake observed in tertiary healthcare settings in South Africa can be attributed to several interconnected factors. Firstly, tertiary facilities often ensure comprehensive access to modern contraceptive methods, including injectables, implants, and intrauterine devices, addressing the availability challenges common in primary care and rural settings [13]. In these facilities, patients benefit from detailed contraceptive counseling and education, fostering informed choices and higher utilization rates [14]. Additionally, healthcare providers in tertiary settings are frequently trained specialists in reproductive health, offering expert guidance and support [13]. Socio-economic variables further contribute, as individuals accessing tertiary care may exhibit higher levels of education and autonomy, both of which are positively correlated with contraceptive use [15]. The enabling legal framework in South Africa, supporting free contraceptive services for individuals over the age of 12, is better implemented in these advanced facilities [14]. Furthermore, cultural and individual empowerment within tertiary settings ensures that women can make autonomous decisions about their reproductive health, unlike other settings where societal norms may pose constraints [15]. These factors collectively distinguish tertiary settings in South Africa, where contraceptive uptake is notably higher than in global and local studies, where barriers such as limited education, stigma, and restricted access remain prevalent.

Historical and structural issues have significantly influenced contraceptive uptake patterns in South Africa. During the Apartheid era, family planning programs were often tied to population control policies, which disproportionately targeted Black communities. This created mistrust in contraceptive services, as they were perceived as tools of oppression rather than empowerment. Additionally, limited access to healthcare infrastructure in rural areas, coupled with cultural and traditional beliefs about fertility, further shaped contraceptive use [16].

Even in the post-Apartheid era, socioeconomic disparities and gender dynamics continue to play a role. Women in low-income communities often face barriers such as inadequate access to healthcare facilities, lack of education about contraceptive options, and stigma surrounding their use. These factors, combined with the legacy of historical injustices, contribute to the complex landscape of contraceptive uptake in South Africa [14].

The proportions of the different family planning methods requested are in keeping with the rest of Sub-Saharan Africa. The DMPA was favored with 48.8% of patients requesting it in our study. In a 2019 systematic review of postpartum contraception use in low- and middle-income countries which included 24 studies, the overall crude pooled estimate of modern contraception use in postpartum women was found to be 41.2%. The most used method was the injectable DMPA, followed by oral contraception and condoms [17].

Only 3.6% of our participants opted for the copper IUCD, which is considered as a first-line method by many organizations globally [18,19]. The Saving Mothers Report in South Africa recommends family planning as part of the national strategy to reduce maternal mortality rates, specifically aiming to improve postpartum long-acting reversable contraception options and improved accessibility to IUCD and the Implanon [20]. These methods are highly effective and do not need to

be taken daily, which leads to improved adherence rates. A drawback of this method is that patients would need to come back six weeks postpartum, which due to financial and social constraints may be difficult and therefore patients are lost to follow up.

The WHO developed the medical eligibility criteria as a guideline for safe contraceptive use. Our results showed that 47.8% of patients received a method that would place them in category three and 3% of patients into category four (Fig 2). The use of the combined oral contraception method in the immediate postpartum period is classified as a category 4 risk, as it poses an unacceptable health risk. Using the COC in the immediate postpartum period increases the risk of clot formation, such as a pulmonary embolism and myocardial infarcts, which can be life threatening [11].

The major contributing factor for most patients to be assigned a MEC category three risk was the patients who received the injectable DMPA. DMPA has been associated to negatively impact breast milk production and thus impact breast feeding [11]. For this reasoning DMPA has been assigned as a category three in the immediate postpartum period. There is also a theoretical risk that high dose progesterone exposure may impact the development of the infant brain. This has been seen in animal studies, but it is unclear if the same impact is seen in humans [11]. The quality of evidence for both of these statements are categorized as low certainty and low grade, which highlights that there is not a direct cause-effect linkage clearly made between these concerns and the use of DMPA. The guideline goes on to acknowledge that in high maternal morbidity and mortality settings with limited access to family planning services, such as low – and lower SES countries, the DMPA may be one of the only methods accessible to postpartum women and thus may be used [9,11]. A systematic review of progesterone-only contraceptive use among breastfeeding women, conducted a year after the release of the WHO MEC document concluded that there is insufficient evidence to demonstrate adverse breastfeeding outcome or negative health outcomes infants. This included restricted growth, health related diseases and impaired development [21].

Interval spacing between pregnancies has been advocated by many organizations worldwide including the WHO as a strategy to lower maternal mortality rate. The majority of our patients had spaced their birth intervals by more than two years, which is in keeping with the recommendation by the WHO of spacing pregnancies more than 24 months apart [4]. Providing adequate family planning has been proven to prolong this period [22]. The unmet need of family planning in Sub-Saharan Africa has led to reports of inadequate spacing by up to 15% [2].

The bivariant and multivariant analysis gave insight into predictive variables for the choice of DMPA over other family planning options. Parity remains the strongest predictor of Depo use. The OR ($0.689$, $p < 0.001$) remains statistically significant, indicating that even after adjustment, higher parity decreases the likelihood of choosing Depo. Women with lower parity often opt for DMPA as a family planning method due to its convenience, effectiveness, and suitability for their reproductive goals. Studies indicate that injectable contraceptives like DMPA are preferred among women who seek long-term, reversible contraception without daily adherence [23]. A lower parity may indicate that there is a desire for future fertility, and this may lead to choosing a reversible method that could easily be stopped when they wish to complete their families.

The bivariant analysis indicated that BMI, HIV status and age were predictors for determining the likelihood of choosing DMPA over other family planning methods. Chronic hypertension (OR $0.881$, $p = 0.535$) and late onset pre-eclampsia (OR $1.200$, $p = 0.216$) was found to have no significant association.

In the multivariant analysis age (OR $1.007$, $p = 0.626$) and HIV status (OR $0.784$, $p = 0.161$) lose their significance in the adjusted model, suggesting that their initial associations were likely confounded by other factors. In the adjusted model BMI (OR $0.986$, $p = 0.053$) was found to have a weak association, indicating it could be a possible influence, with a slight reduction in DMPA use as BMI increases.

There could be possible limitations to our results in terms of information bias as the findings of the study are based on data collected from the maternity records and medical notes, which could have missing or incorrectly recoded information. The information collected was done at discharge, however we do not know if these women continued their family planning method or breastfeeding after discharge, as their follow up appointments were not included in this study. Incorporating

postpartum follow-up data could potentially have enhanced the study by providing additional insights into the effects of DMPA on breastfeeding and the continuation rates of selected family planning methods. However, the feasibility of such data collection was limited, as the majority of patients receive follow-up care at community clinics across the province rather than returning to Tygerberg Academic Hospital. This decentralized follow-up pattern presents logistical challenges and may have contributed to anticipated attrition, making consistent data collection beyond the initial postpartum period difficult within the scope of this study.

## Conclusion

The exceptionally high uptake of family planning indicates it can be successfully implemented in the postpartum period. Parity appears to be the strongest predictor of choosing DMPA over other family planning methods.

Individualized counselling remains essential to support informed decision-making regarding contraceptive methods, with careful consideration of long-term use and the mitigation of potential adverse outcomes. Health care professionals should enhance awareness of long-acting reversible contraception, as endorsed by the WHO, while also acknowledging and respecting patient preferences. In contexts where breastfeeding is practiced, it is important that family planning approaches are selected to align with and support the continuation of lactation and provide the safest and most appropriate option for the patient.

## Supporting information

**S1 File. The supplementary file is the raw data set of this data collection.**
(XLSX)

## Author contributions

**Conceptualization:** Denise Sproul, Gerhard Hanekom, Michael McCaul.

**Data curation:** Denise Sproul.

**Formal analysis:** Denise Sproul.

**Funding acquisition:** Denise Sproul.

**Investigation:** Denise Sproul.

**Methodology:** Denise Sproul, Michael McCaul.

**Project administration:** Denise Sproul.

**Supervision:** Gerhard Hanekom, Michael McCaul.

**Visualization:** Denise Sproul.

**Writing – original draft:** Denise Sproul.

**Writing – review & editing:** Denise Sproul, Gerhard Hanekom, Michael McCaul.

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
