## [Decision Letter · Decision Letter 0]

15 Jan 2025

PONE-D-24-39098A cross-sectional prevalence study demonstrating a high uptake of postpartum family planning in a high risk setting in South Africa.PLOS ONE

Dear Dr. Sproul,

Thank you for submitting your manuscript to PLOS ONE. After careful consideration, we feel that it has merit but does not fully meet PLOS ONE’s publication criteria as it currently stands. Therefore, we invite you to submit a revised version of the manuscript that addresses the points raised during the review process.

**ACADEMIC EDITOR:**  In addition to addressing the comments of the reviewers, authors should address the following concerns1.Title: Authors should remove the academic qualifications from the authors' list The title needs review. "A cross-sectional prevalence study demonstrating a high uptake of postpartum family planning in a high risk setting in South Africa." what "high risk" are the authors referring to? HIV prevalence? poor contraceptive uptake? etc. I suggest: "Prevalence and sociodemographic risk and morbidities associated with contraceptive uptake among postpartum women at ...... A cross-sectional study". This is a suggestion that can be reviewed by the authors.2.Authors should use continuous number line when submitting for efficient review3. Authors should read through the authors guideline and adhere strictly to it. 4. Authors should add more numbers to the statistical result of the abstract.5.Please remove the Table in the introduction6.The statistical analysis did not describe a rigorous statistical analysis that is of quality for publishing in Plos one. I strongly recommend that authors should conduct bivariate analysis and if possible appropriate regression modelling to before the manuscript with be of high quality that can be published in PLOS ONE. The sample size is adequate for useful conclusion from the study7. Authors stated .".....standard deviation (SD), and range for continuous variables if normally distributed" What about if the continuous variable was not normally distributed?8.In the Result section, authors stated "Over a period of seven months 1066 participants were recruited for this study and had a recorded family planning method of their choice."  From the methodology section, it appears that the authors described retrospective chart review of routinely collected data. Please authors should be explicit about what study was conducted. was it a retrospective or prospective study?9Table 1 is not looking "professionsl" For example: "Delivery Characteristics" lumped many unrelated categories. One of the variables in that category is mode of delivery that can be vaginal delivery, operative vaginal delivery or caesarean section. Birth outcome such as live birth, fresh still birth, early neonatal death etc may be another category etc,10. This study seems to be among postpartum women. When after delivery was the interview conducted.? If it was anytime post partum. It will be good to know how soon after delivery that women presented for family planning at the hospitals. 11. Table 2.  gravidity and parity are not likely to be normally distributed. Thus median and IQR may be more appropriate. Please check12.Disucssion: Authors to cite relevant literatures as required13. Please ensure that DMPA had been previously defined in the manuscript14. Are there any historical and structural issues that influences the contraceptive uptake pattern? . 15, Social cultural and economic determinants of health is a very important issue  in South Africa. However, authors did not describe the ethnic characteristics of the studied participants. Occupation and income of the participants are also important information to provide. 16. If the prevalence is so high, is it possible to collect further data for more months to assist to obtain more participants that will not take the contraceptive for easy comparative cross-sectional study?17. What are the reasons for the comparatively high contraceptive uptake in this population as compared with other south African and global studies?18. Authors stated "There is also a theoretical risk that high dose progesterone exposure may impact the development of the infant brain. Authors should cite appropriate reference for this statement. Please ensure that your decision is justified on PLOS ONE’s publication criteria and not, for example, on novelty or perceived impact.

We look forward to receiving your revised manuscript.

Kind regards,

Gbenga Olorunfemi, MBBS,MSC,FMCOG,FWASC

Academic Editor

PLOS ONE

Journal Requirements:

3. Thank you for stating the following financial disclosure: We would like to acknowledge the Temporary Research funding support from Stellenbosch University’s Subcommittee C of the Research Committee. As well as the funding received from the GELA project which is part of the EDCTP2 programme supported by the European Union (grant number RI2020S-3303 - GELA). The views and opinions of authors expressed herein do not necessarily state or reflect those of EDCTP.

Reviewers' comments:

Reviewer's Responses to Questions

**Comments to the Author**

1. Is the manuscript technically sound, and do the data support the conclusions?

Reviewer #1: Yes

Reviewer #2: Yes

2. Has the statistical analysis been performed appropriately and rigorously? 

Reviewer #1: Yes

Reviewer #2: Yes

3. Have the authors made all data underlying the findings in their manuscript fully available?

Reviewer #1: Yes

Reviewer #2: Yes

4. Is the manuscript presented in an intelligible fashion and written in standard English?

Reviewer #1: Yes

Reviewer #2: Yes

5. Review Comments to the Author

Reviewer #1: 1.The introduction and methodology are apt and clearly stated the justification for the study and objectives.

2.The authors followed sound ethical practice and also declared the funding for the study.

3.The Discussion followed sound scientific principles.

However,

1. Table 1 seems o be an unnecessary inclusion having referenced it in the manuscript and is readily available for access online. If the table must be included, it should be in the methodology section since WHO MEC was used for assessment.

2. The use of Participants when data used is secondary seems odd as information/ data was extracted from records already available. This is more so having already stated the data inclusion criteria

Reviewer #2: STRENGTHS

The study's statistical analysis exhibited several strengths, including a well-aligned cross-sectional design that effectively determined the prevalence of family planning uptake, and a large sample size that ensured adequate statistical power. The use of frequency, percentages, and 95% confidence intervals provided robust prevalence estimates, underscoring the study's success in capturing immediate postpartum contraceptive uptake.

The study effectively summarized demographics using descriptive statistics. The Pregnancy Eligibility Criteria framework added a clinical dimension. The study appropriately highlighted the dominance of injectable DMPA, spacing trends, supported WHO recommendations, and contextualized the findings against regional data to underscore the high uptake rate compared to previous studies.

LIMITATIONS

The absence of multivariate models restricted exploration of independent predictors for family planning uptake and method choice. Confounding variables like socioeconomic status, maternal age, and comorbidities could have influenced outcomes. Furthermore, the reliance on discharge records without follow-up data introduced a time-dependent bias, as the study could not assess long-term adherence to contraceptive methods. Advanced techniques like survival analysis could have enriched the investigation of interpregnancy intervals.

In conclusion, the study’s findings provide valuable insights into postpartum contraceptive uptake in a high-risk population. Strengthening the analysis with multivariate modeling and advanced statistical methods would enhance the study’s rigor and applicability.

TYPOGRAPHICAL ERRORS

Pg 8: delete full stop at the end of the title

Pg 14: (Statistical Analysis) correction “confident interval” for “confidence interval”

Pg 15: (Results) correction “summarized” for “summarized” consistency in British spelling

6. PLOS authors have the option to publish the peer review history of their article (what does this mean?). If published, this will include your full peer review and any attached files.

Reviewer #1: No

Reviewer #2: **Yes:** Ogelle, Onyecherelam Monday

---

## [Author Response · Author response to Decision Letter 1]

10 Apr 2025

Changes requested Revision action.

Title: Authors should remove the academic qualifications from the authors' list The title needs review. "A cross-sectional prevalence study demonstrating a high uptake of postpartum family planning in a high risk setting in South Africa." what "high risk" are the authors referring to? HIV prevalence? poor contraceptive uptake? etc. I suggest: "Prevalence and sociodemographic risk and morbidities associated with contraceptive uptake among postpartum women at ...... A cross-sectional study". This is a suggestion that can be reviewed by the authors. The title has been revised and edited accordingly. The authors academic qualifications have been removed.

Authors should use continuous number line when submitting for efficient review Continuous number line have been inserted.

PLOS requires an ORCID iD for the corresponding author in Editorial Manager on papers submitted after December 6th, 2016. Please ensure that you have an ORCID iD and that it is validated in Editorial Manager. To do this, go to ‘Update my Information’ (in the upper left-hand corner of the main menu), and click on the Fetch/Validate link next to the ORCID field. This will take you to the ORCID site and allow you to create a new iD or authenticate a pre-existing iD in Editorial Manager. Orchid numbers have been added and updated.

Denise Sproul: 0009-0003-2138-0495

Gerhard Hanekom: 0000-0002-2832-9436

Michael McCaul: 0000-0002-2730-6478

Thank you for stating the following financial disclosure: We would like to acknowledge the Temporary Research funding support from Stellenbosch University’s Subcommittee C of the Research Committee. As well as the funding received from the GELA project which is part of the EDCTP2 programme supported by the European Union (grant number RI2020S-3303 - GELA). The views and opinions of authors expressed herein do not necessarily state or reflect those of EDCTP.

Please include this amended Role of Funder statement in your cover letter; we will change the online submission form on your behalf All related revisions have been updated, and additional statement has been added to the manuscript and cover letter.

TYPOGRAPHICAL ERRORS

Pg 8: delete full stop at the end of the title

Pg 14: (Statistical Analysis) correction “confident interval” for “confidence interval”

Pg 15: (Results) correction “summarized” for “summarized” consistency in British spelling

All correctios have been made.

3. Authors should read through the authors guideline and adhere strictly to it. We have made adjustments accordingly.

4. Authors should add more numbers to the statistical result of the abstract. More results have been added to the abstract section and shown in red.

5.Please remove the Table in the introduction The table has been removed as requested.

6.The statistical analysis did not describe a rigorous statistical analysis that is of quality for publishing in Plos one. I strongly recommend that authors should conduct bivariate analysis and if possible appropriate regression modelling to before the manuscript with be of high quality that can be published in PLOS ONE. The sample size is adequate for useful conclusion from the study A bivariant analysis and multivariant analysis have been conducted and findings have been added to relevant sections.

Authors stated .".....standard deviation (SD), and range for continuous variables if normally distributed" What about if the continuous variable was not normally distributed? This has been revised and reviewed in the manuscript. If not normally distributed interquartile ranges and medians have been used.

In the Result section, authors stated "Over a period of seven months 1066 participants were recruited for this study and had a recorded family planning method of their choice." From the methodology section, it appears that the authors described retrospective chart review of routinely collected data. Please authors should be explicit about what study was conducted. was it a retrospective or prospective study? A prospective study was conducted. This has been revised in the manuscript.

Table 1 is not looking "professional" For example: "Delivery Characteristics" lumped many unrelated categories. One of the variables in that category is mode of delivery that can be vaginal delivery, operative vaginal delivery or caesarean section. Birth outcome such as live birth, fresh still birth, early neonatal death etc may be another category etc, The tables have been split and edited as requested into table 2 and 3.

This study seems to be among postpartum women. When after delivery was the interview conducted.? If it was anytime post partum. It will be good to know how soon after delivery that women presented for family planning at the hospitals. The data was collected from the medical records on the day of discharge after delivery. This has been stated in the methods section as well.

Table 2. gravidity and parity are not likely to be normally distributed. Thus median and IQR may be more appropriate. Please check These statistics have been re-analysed and updated accordingly.

Discussion: Authors to cite relevant literatures as required Revisions done.

Please ensure that DMPA had been previously defined in the manuscript DMPA has been defined clearly in the text.

Are there any historical and structural issues that influences the contraceptive uptake pattern? Additional paragraph added to discussion as requested.

Social cultural and economic determinants of health is a very important issue in South Africa. However, authors did not describe the ethnic characteristics of the studied participants. Occupation and income of the participants are also important information to provide. This revision is addressed and discussed in the discussion section.

If the prevalence is so high, is it possible to collect further data for more months to assist to obtain more participants that will not take the contraceptive for easy comparative cross-sectional study? This limitation is addressed in the limitations section.

What are the reasons for the comparatively high contraceptive uptake in this population as compared with other south African and global studies? Revised and discussed in the discussion section.

Authors stated "There is also a theoretical risk that high dose progesterone exposure may impact the development of the infant brain. Authors should cite appropriate reference for this statement The WHO statement that makes this claim is cited as well as the original data.

---

## [Decision Letter · Decision Letter 1]

6 Jul 2025

PONE-D-24-39098R1

Prevalence and sociodemographic risk and morbidities associated with contraceptive uptake among postpartum women at Tygerberg Academic Hospital, Cape Town: A cross-sectional study.

PLOS ONE

Dear Dr. Sproul,

Thank you for submitting your manuscript to PLOS ONE. After careful consideration, we feel that it has merit but does not fully meet PLOS ONE’s publication criteria as it currently stands. Therefore, we invite you to submit a revised version of the manuscript that addresses the points raised during the review process.

We look forward to receiving your revised manuscript.

Kind regards,

Dubale Dulla Koboto, MSc

Academic Editor

PLOS ONE

Reviewers' comments:

Reviewer's Responses to Questions

**Comments to the Author**

1. If the authors have adequately addressed your comments raised in a previous round of review and you feel that this manuscript is now acceptable for publication, you may indicate that here to bypass the “Comments to the Author” section, enter your conflict of interest statement in the “Confidential to Editor” section, and submit your "Accept" recommendation.

Reviewer #1: (No Response)

2. Is the manuscript technically sound, and do the data support the conclusions?

Reviewer #1: Yes

3. Has the statistical analysis been performed appropriately and rigorously? 

Reviewer #1: Yes

4. Have the authors made all data underlying the findings in their manuscript fully available?

Reviewer #1: Yes

5. Is the manuscript presented in an intelligible fashion and written in standard English?

Reviewer #1: No

6. Review Comments to the Author

Reviewer #1: Thank you for requesting my opinion of this manuscript.

The importance of postpartum contraception in maternal morbidity and mortality reduction is well documented. However I have the following observations:

1. Title: While I agree with Prevalence as being the main objective, `Sociodemographic risk and morbidity associated with contraception` was not studied, but rather as part of the eligibility criteria. I suggest that the title be reviewed.

2. Aim: The aim, lines 73-78 should be rephrased. Line 78 seems presumptuous and and conclusive before the study.

3. Methodology: The study cannot be cross-sectional and prospective at the same time as stated by the researchers. The data could as well have be collected retrospectively from January to July 2023. The methodology should be revised. The WHO eligibility criteria being a readily available document as referenced(Table 1) need not be included.

4. Conclusions: This need rephrasing . Lines 341-348 cannot be deduced and recommendation made as such.

5. Generally: Mixed British and American English abound in the write-up, eg Organization, recognised, individualized in lines 59, 51, 68 as examples. writer can crosscheck and provide uniformity.

Other observations are in the suggestion in the attached as annotation.

Above are my suggestions.

Thank you

Dr. Abah MG

7. PLOS authors have the option to publish the peer review history of their article (what does this mean?). If published, this will include your full peer review and any attached files.

Reviewer #1: No

---

## [Author Response · Author response to Decision Letter 2]

15 Jul 2025

15 July 2025

Dear Editor in Chief,

Thank you for the opportunity to re-submit our article to PLOS ONE journal. Please find below a rebuttal table outlining the changes and revisions requested for our submission.

Changes requested Revision action.

1. Title: While I agree with Prevalence as being the main objective, `Sociodemographic risk and morbidity associated with contraception` was not studied, but rather as part of the eligibility criteria. I suggest that the title be reviewed. Purposed new title: “Contraceptive Uptake in the Postpartum Period: Prevalence Among Women Attending Tygerberg Academic Hospital.”

2. Aim: The aim, lines 73-78 should be rephrased. Line 78 seems presumptuous and and conclusive before the study. This paragraph has been revised. See corrections and revisions in text.

3. Methodology: The study cannot be cross-sectional and prospective at the same time as stated by the researchers. The data could as well have be collected retrospectively from January to July 2023. The methodology should be revised. The WHO eligibility criteria being a readily available document as referenced(Table 1) need not be included. Thank you for your opinion. We have changed the study method to a descriptive study design to be more inclusive of the prospective data collection interval.

I have deleted the WHO table 1. All table numbers have been updated to follow sequence of numbers.

4. Conclusions: This need rephrasing . Lines 341-348 cannot be deduced and recommendation made as such. This paragraph has been revised. See corrections and revisions in text.

5. Generally: Mixed British and American English abound in the write-up, eg Organization, recognised, individualized in lines 59, 51, 68 as examples. writer can crosscheck and provide uniformity. Grammer, spelling formatting check have been done to entire manuscript.

All revisions and changes have also been marked in the text using red font and comment boxes.

Kind regards

Denise Sproul

---

## [Editor Report · Decision Letter 2]

20 Mar 2026

PONE-D-24-39098R2Contraceptive Uptake in the Postpartum Period: Prevalence Among Women Attending Tygerberg Academic HospitalPLOS One

Dear Dr. Sproul,

Thank you for submitting your manuscript to PLOS ONE. After careful consideration, we feel that it has merit but does not fully meet PLOS ONE’s publication criteria as it currently stands. Therefore, we invite you to submit a revised version of the manuscript that addresses the points raised during the review process.

We look forward to receiving your revised manuscript.

Kind regards,

Alejandro Torrado Pacheco, PhD

Associate Editor

PLOS One

on behalf of

Alfredo Luis Fort, M.D., M.Sc., Ph.D.

Academic Editor

PLOS One

Journal Requirements:

Additional Editor Comments:

This is the 2nd Revision for this manuscript. Although it has been importantly improved, there are several places where better descriptions would mean better understanding to readers. Also, the most important change that I consider needs to be done is not to use the word "Prevalence" since the numbers found in the study do not coincide with the definition of prevalence as representative within the larger community. I am taking the decision to accept the manuscript but right after you the authors have made the necessary small but important changes I suggest in the attached file. Thanks!

---

## [Author Response · Author response to Decision Letter 3]

19 Apr 2026

Thank you for the revisions and comments. I have addressed each revision and outlined those details in my letter of response to the reviewers.

---

## [Editor Report · Decision Letter 3]

5 May 2026

Contraceptive Uptake and Characteristics in the Immediate Postpartum Period Among Women Attending Tygerberg Academic Hospital

PONE-D-24-39098R3

Dear Dr. Sproul,

We’re pleased to inform you that your manuscript has been judged scientifically suitable for publication and will be formally accepted for publication once it meets all outstanding technical requirements.

Kind regards,

Alfredo Luis Fort, M.D., M.Sc., Ph.D.

Academic Editor

PLOS One

Additional Editor Comments (optional):

The authors made great efforts to improve the several inconsistencies and some writing that needed improvement. So, it is now ready for publication (minor edits may be needed but that's OK during the publication process). Well done.

---

## [Editor Report · Acceptance letter]

PONE-D-24-39098R3

PLOS One

Dear Dr. Sproul,

I'm pleased to inform you that your manuscript has been deemed suitable for publication in PLOS One. Congratulations! Your manuscript is now being handed over to our production team.

Kind regards,

on behalf of

Dr. Alfredo Luis Fort

Academic Editor

PLOS One